# Impact of calcined bauxite aggregates on the polishing resistance and skid resistance performance of SMA-7 asphalt mixtures

**Pengfei Li[1], Lingkun Kong[1], Nan Mao[2,3], Chenwei Gu [2,3]***

**1** CCCC Second Highway Engineering Bureau Third Engineering CO., LTD, Xi'an, China, **2** School of highway, Chang'an University, Xi'an, China, **3** Key Laboratory for Special Area Highway Engineering of Ministry of Education, Chang'an University, Xi'an, China

\* 2020021051@chd.edu.cn

## Abstract

High-performance anti-skid asphalt mixtures are essential for improving skid resistance and pavement durability. This study investigates the skid resistance performance of small-aggregate-size SMA-7 asphalt mixtures using calcined bauxite (CB) aggregates. Four types of aggregates—75#, 85#, and 88# CB and limestone—were used in the mixture preparation. Various laboratory tests, including pavement performance, Polished Stone Value (PSV) of aggregates, three-wheel polishing, and dynamic friction tests, were conducted to evaluate the performance and friction characteristics of the mixtures at various polishing stages. The results indicate that the optimal coarse-to-fine aggregate ratio for SMA-7 is 75:25, with a maximum particle size of 6.35 mm. The PSV of 88# CB aggregate stabilizes after 120,000 polishing cycles, exhibiting a decay rate 30% slower than that of limestone aggregates. Among the mixtures, 88#CB-SMA demonstrates superior high-temperature stability (1.5 times higher than limestone), slightly better low-temperature crack resistance, and significantly enhanced polishing resistance. Additionally, the dynamic friction coefficients of CB mixtures show slower attenuation, retaining 29.4–36.3% higher residual friction compared to limestone even after prolonged polishing. Strong correlations ($R^2 > 0.85$) between the attenuation rates of PSV and dynamic friction coefficients confirm that the enhanced wear resistance of CB aggregates is key to long-term skid resistance, particularly at lower speeds. These findings suggest that high-grade CB aggregates greatly improve both the skid resistance and overall performance of asphalt mixtures, providing valuable insights for designing durable small-size asphalt wear layers.

**Data availability statement:** All relevant data are within the manuscript and its Supporting Information files.

**Funding:** This research was supported by the fundamental research funds for the central universities, CHD (300102213509).

**Competing interests:** The authors have declared that no competing interests exist.

## 1 Introduction

Asphalt pavement skid resistance plays a critical role in ensuring road safety, particularly under heavy traffic and adverse weather conditions [1,2]. While newly constructed pavements typically meet initial skid resistance standards, prolonged traffic polishing and environmental aging lead to progressive deterioration of surface textures, often resulting in pavements that can only meet the critical traffic safety requirements at the end of design life [3–5]. Recent advances in material durability analysis, demonstrate that degradation kinetics are governed by multi-scale damage mechanisms spanning from micro-defect nucleation to macro-performance decay [6–8]. For asphalt mixtures, this degradation is governed by two interrelated factors: aggregate properties and mixture gradation, which has emerged as the dominant determinant of long-term skid retention [9].

Stone matrix asphalt (SMA) pavements offer advantages such as high density, noise reduction, and durability. Compared with Asphalt Concrete (AC) pavements, SMA provides better texture and a larger contact area with tires, resulting in superior skid resistance, making it well-suited for use as a skid-resistant wearing course [10,11]. Research by the National Center for Asphalt Technology (NCAT) has demonstrated that SMA-5, with a nominal maximum aggregate size of 4.75 mm, offers excellent skid and rutting resistance, leading to its widespread use in ultra-thin wearing courses across the U.S. [12,13]. However, the small particle size of SMA-5 compromises construction feasibility and long-term texture retention compared to larger NMAS mixtures like SMA-10 [14,15]. For instance, Liu et al. found that SMA-10 achieves significantly higher British Pendulum Number (BPN) values and average texture depth due to its coarser aggregate structure [16].

Despite these advantages, both SMA-5 and SMA-10 have inherent limitations: small particle size of SMA-5 limits its long-term skid resistance, while larger aggregates pose challenges in the construction of ultra-thin wearing course [17]. These drawbacks highlight the need for a balanced approach that optimizes texture depth, construction feasibility, and long-term durability. Developing an ultra-thin wearing course suitable for diverse traffic environments has thus become a key research focus, with particular emphasis on optimizing aggregate properties and gradation design.

The skid resistance of asphalt pavements depends not only on gradation but also fundamentally on aggregate properties [18–20], where dual-scale texture mechanisms govern frictional behavior [21]. At the micro-scale, mineral composition determines surface asperity density and resistance to polishing-induced smoothing [22,23]. At the macro-scale, particle geometry controls drainage capacity through void structure stability [24]. Aggregates subjected to traffic polishing undergo progressive degradation of both texture dimensions, collectively diminishing friction retention over time. This degradation process is visually captured in comparative studies of aggregate surfaces under wear simulation [25]. Aggregates' ability to maintain functional texture characteristics, which is quantified by the polished stone value (PSV) for micro-roughness and mean profile depth (MPD) for macro-structure, directly governs

long-term friction durability [26]. Recent studies highlight that optimal aggregate selection must balance vertical load-bearing capacity with horizontal abrasion resistance under traffic shear forces [27], requiring materials that simultaneously preserve both texture scales under prolonged mechanical stress.

Calcined bauxite (CB), as a new high-hardness skid-resistant aggregate, has superior polishing and wear resistance compared to traditional aggregates [28–30]. Its alumina-dominated microstructure demonstrates exceptional stability against chemical degradation, analogous to sulfuric acid-resistant concrete incorporating coal waste aggregates [31,32]. Laboratory testing has proven the adhesive properties between asphalt and CB aggregates [33]. Meanwhile, studies have found that the primary wear-resistant components of calcined bauxite are corundum and mullite. The significant hardness difference between these two minerals gives calcined bauxite excellent long-term wear resistance and polishing performance [34,35]. Li et al. analyzed the chemical composition of calcined bauxite and found that its incorporation significantly improved the skid resistance of asphalt mixtures [36]. Research by Guan et al. showed that calcined bauxite exhibits high skid resistance in the polishing process, making it suitable for high-friction surface treatments [37,38]. However, existing research predominantly focuses on standard-thickness layers, leaving a gap in understanding CB's performance in ultra-thin wearing courses, where aggregate size and interlock dynamics differ significantly.

In China, despite CB's proven advantages, its adoption in ultra-thin pavements remains limited, with scarce studies addressing its long-term polishing behavior and skid resistance decay patterns. To bridge this gap, this study investigates four aggregates (75#, 85#, 88# CB, and LS) within a novel SMA-7 framework optimized via multi-point skeleton theory and V-S design methods. The mixtures' mechanical performance (high/low-temperature stability, moisture resistance) and skid durability were rigorously tested. Accelerated polishing and three-wheel abrasion experiments were conducted to quantify PSV attenuation and dynamic friction coefficient decay in terms of aggregate properties. Additionally, the macro-texture depth of the mixtures was assessed, providing insights into the factors affecting the skid resistance of the mixtures. These findings will contribute valuable knowledge to the design of high skid-resistant ultra-thin surface layers.

## 2 Materials and methods

### 2.1 Materials

**2.1.1 Asphalt.** SBS I-D Modified asphalt was used as a binder, and its technical properties are listed in Table 1.

**2.1.2 Aggregate.** The primary objective of this study is to investigate the influence of different types of aggregates on skid resistance. Four types of aggregates were used in the experiment: three grades of CB (75#, 85#, and 88#) and LS (as the control group). The key mechanical and physical properties of these aggregates are summarized in Table 2, while their chemical compositions are detailed in Tables 3. The CB grades were classified based on their $Al_2O_3$ content, with higher-grade CB having higher $Al_2O_3$ content, hardness, and polished stone value (PSV), along with a lower Los Angeles abrasion (LAA), contributing to superior physical and mechanical performance.

Table 4 presents the mineralogical composition and detailed hardness metrics of the aggregates. The Aggregate Hardness Parameter is derived from the mineral composition and their corresponding Mohs hardness values, providing an

**Table 1. Main technical specifications of modified asphalt.**

| Test indicators | | Value | Unit | Specification |
|---|---|---|---|---|
| Penetration at 25 °C | | 58 | 0.1mm | ASTM D5-97 |
| Ductility at 5 °C | | 37 | cm | ASTM D113-99 |
| Softening point (R&B) | | 61.0 | °C | ASTM D36-06 |
| Specific gravity | | 1.024 | — | ASTM D70-76 |
| RTFOT (163°C,75min) | Mass loss | −0.13 | % | ASTM D2872-04 |
| | Penetration ratio at 25 °C | 76.0 | % | ASTM D5-97–06 |
| | Ductility at 5 °C | 19.0 | cm | ASTM D113-99 |

**Table 2. Mechanical and physical properties of aggregate.**

| Test items | Value | | | | Unit |
|---|---|---|---|---|---|
| | LS | 75#CB | 85#CB | 88#CB | |
| Apparent relative density | 2.860 | 3.259 | 3.435 | 3.490 | g·cm⁻³ |
| Relative density in gross volume | 2.765 | 2.743 | 3.181 | 3.218 | — |
| Water absorption | 1.300 | 5.818 | 4.503 | 4.300 | % |
| Crush value | 22.71 | 0.65 | 0.35 | 7.76 | % |
| Flat and elongated particle content | 3.85 | 18.67 | 8.52 | 0.44 | % |
| Los Angeles abrasion loss | 21.70 | 13.82 | 10.99 | 10.63 | % |
| Polished stone value | 38.4 | 45.7 | 53.8 | 55.3 | BPN |

**Table 3. Chemical composition of aggregates.**

| Type | $SiO_2$ | $TiO_2$ | $Al_2O_3$ | $Fe_2O_3$ | MgO | CaO | $Na_2O$ | $K_2O$ | $P_2O_5$ | Other |
|---|---|---|---|---|---|---|---|---|---|---|
| LS | 14.6 | 3.08 | 2.30 | 1.70 | 2.44 | 54.57 | 3.63 | 1.86 | 2.16 | 15.66 |
| 75#CB | 20.31 | 2.32 | 75.53 | 1.45 | 0.22 | 0.17 | 0.03 | 0.14 | 0.18 | 0.65 |
| 85#CB | 10.18 | 2.45 | 85.24 | 1.37 | 0.19 | 0.15 | 0.02 | 0.18 | 0.22 | 1.08 |
| 88#CB | 3.32 | 4.47 | 90.29 | 1.55 | 0.15 | 0.17 | 0.01 | 0.17 | 0.24 | 1.63 |

**Table 4. Main mineral parameters of aggregates.**

| Type | Main mineral type | Composition | Moh's hardness | Aggregate Hardness Parameter |
|---|---|---|---|---|
| LS | Calcite | 71.4 | 4.0 | 4.98 |
| | Dolomite | 28.6 | 3.5 | |
| 75#CB | Corundum | 76.0 | 9.0 | 9.78 |
| | Mullite | 24.0 | 6.0 | |
| 85#CB | Corundum | 84.0 | 9.0 | 9.89 |
| | Mullite | 16.0 | 6.0 | |
| 88#CB | Corundum | 87.0 | 9.0 | 10.08 |
| | Mullite | 13.0 | 6.0 | |

assessment of the hardness of each individual aggregate [4]. The table further highlights that CB aggregates are primarily composed of corundum and mullite, both of which exhibit high Mohs hardness values. This superior hardness contributes to the overall strength of the aggregates, making CB a promising candidate for high-skid-resistance applications. The macroscopic and microscopic surface textures of the aggregates are illustrated in Fig 1.

In addition, mineral powder plays a key role in asphalt mixtures. The addition of mineral powder interacts with asphalt, significantly reducing the phase angle of the asphalt binder. In this study, limestone powder and lignin fiber were selected for the mixture.

## 2.2 Experiment method

### 2.2.1 Pavement performance testing.
The asphalt mixtures, prepared with various aggregate types and optimized asphalt-aggregate ratios, underwent a series of performance tests to evaluate their high-temperature stability, low-temperature crack resistance, and water stability. These tests were conducted in accordance with China standard "Test Specifications for Asphalt and Asphalt Mixtures in Highway Engineering" (JTG E20-2011). Specifically, the mixtures were subjected to high-temperature rutting tests, low-temperature bending tests, water immersion Marshall tests, and freeze-thaw splitting tests.

### 2.2.2 PSV attenuation test.

The PSV attenuation test was conducted to evaluate the changes in the PSV of aggregates over multiple polishing cycles. The test was performed in accordance with the China standard "Test Methods of Aggregates for Highway Engineering" (JTG 3432−2024), with the basic procedure illustrated in Fig 2.

Four types of aggregates were prepared into test specimens after sieving and washing. Aggregate particles within the nominal size range of 4.75–9.5 mm were selected for specimen fabrication. The aggregates were densely packed in a mold, with fine sand filling the interstitial spaces. Epoxy resin was used as a binder to ensure the aggregates were securely adhered within the mold. The specimens were then cured at 40°C for 3 hours, followed by 9 hours of natural cooling before demolding. Four replicate specimens were prepared for each type of aggregate.

The specimens were subjected to accelerated polishing using a rubber-wheel polishing machine, which simulated the load action of road traffic under continuous water flow carrying abrasive grit. The loading cycles were set from 0 to 270,000 times, with an interval of 30,000 cycles. At each interval, the specimens were removed from the machine, rinsed to remove polishing residue, and air-dried at room temperature. The PSV of each specimen was measured using a pointer-type pendulum friction coefficient tester, with the results expressed in BPN. This process was repeated until all

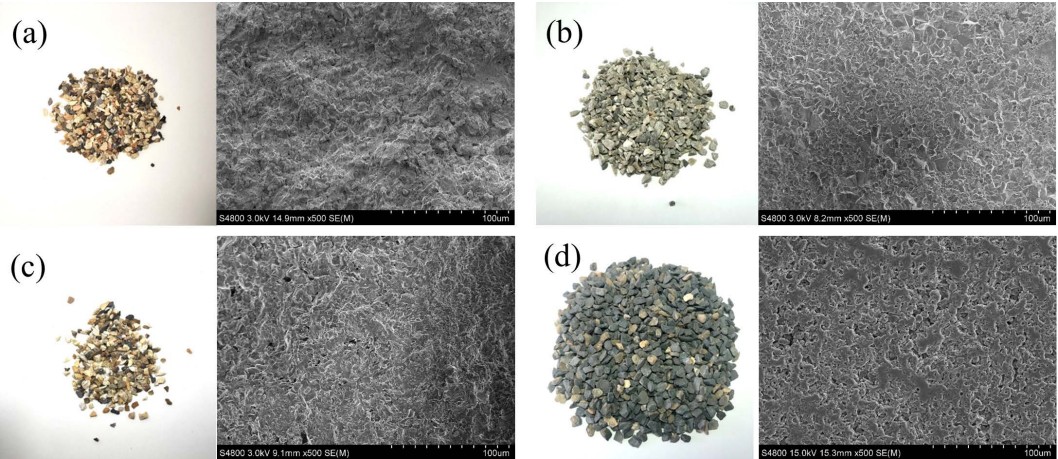

**Fig 1. Macroscopic morphology and SEM images.** (a) LS; (b) 75# CB; (c) 85# CB; (d) 88# CB.

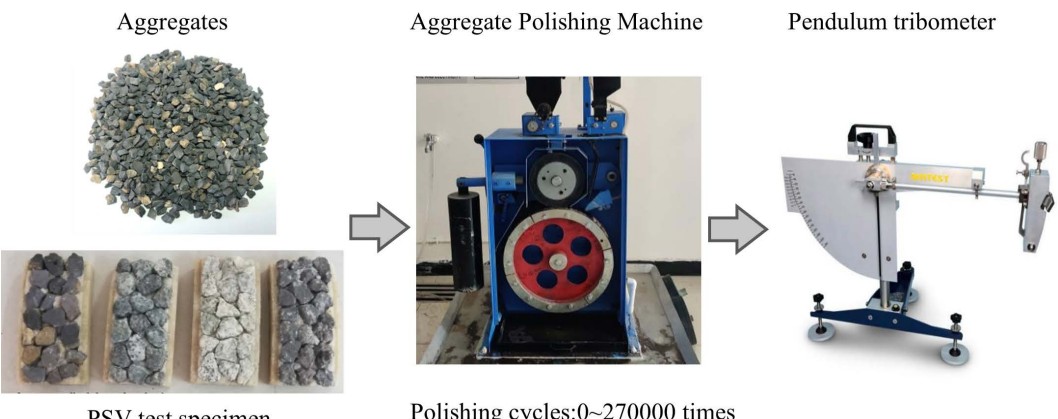

Aggregates          Aggregate Polishing Machine          Pendulum tribometer

PSV test specimen          Polishing cycles:0~270000 times

**Fig 2. PSV test process.**

testing cycles were completed. The final PSV value for each aggregate type at each interval was calculated as the average of four replicate specimens.

**2.2.3 Accelerated wear and dynamic friction test (DFT).** The PSV test was conducted for the aggregates. For asphalt mixtures, an accelerated wear test was performed using a three-wheel accelerated wear testing device, as shown in Fig 3. The device was configured with a preset load, ensuring automatic pressure maintenance throughout the testing process. The tire pressure was set at 0.75 MPa, and the wheel track diameter on the specimen was precisely 284 mm, which corresponds to the dimensions used in the DFT wheel track.

During the test, three rubber tires simulate vehicle tire movement over the asphalt pavement. The wear cycles were carried out progressively, stopping at 2,500, 5,000, 10,000, 20,000, 30,000, and 50,000 cycles to assess the wear resistance.

To evaluate the friction performance under the wear test, the dynamic friction coefficient was measured using a JDF-08 dynamic friction coefficient tester (Fig 3c), with testing speeds set at 20~80 km/h, with 10 km/h as the interval. For each type of mixture and at each testing speed, five specimens were prepared(4×5×7). The corresponding friction coefficients were denoted as µ20~µ80 for each respective speed.

**2.2.4 MPD test.** This study employs the AMES road laser texture scanner to measure the MPD of the specimens. The MPD value reflects the surface texture parameters of the pavement, thereby indirectly indicating the road's anti-slip and wear resistance capabilities. Fig 4 illustrates the testing equipment and measurement process. By scanning the road surface before and after wear using the AMES laser scanner, the roughness of the road surface can be assessed and the MPD value can be calculated.

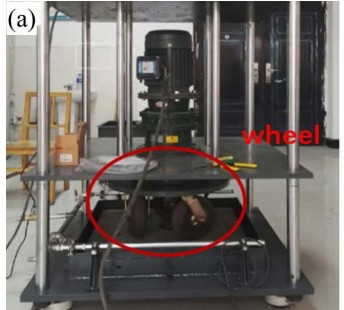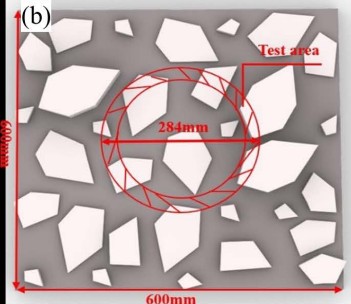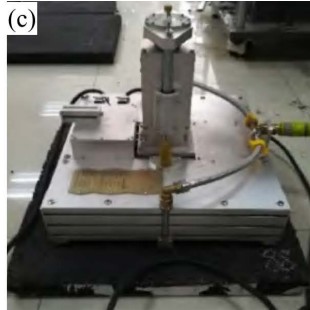

**Fig 3. Device for DFT.** (a) Three-wheel accelerated wear test; (b) specimen layout and test area; (c) JDF-08 dynamic friction coefficient tester.

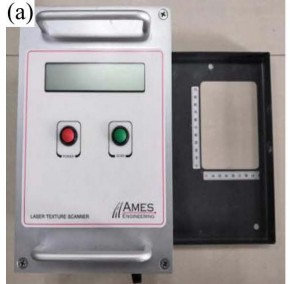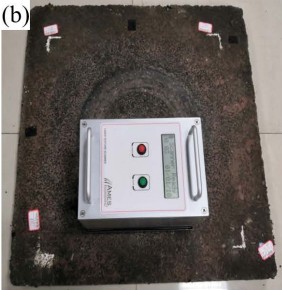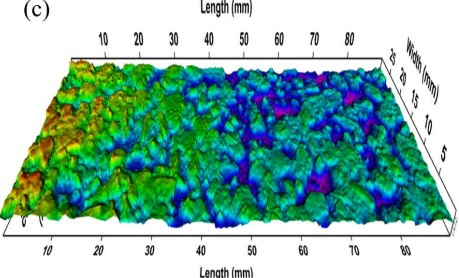

**Fig 4. MPD test.** (a) AMES scanner; (b) Measurement of the specimen; (c) 3D image for MPD.

## 2.3 SMA-7 gradation composition design and optimization

**2.3.1 Redefinition of coarse and fine aggregate boundaries for SMA-7 mixtures.** Based on ASTM aggregate particle size requirements, a new sieve size of 6.35 mm (1/4 inch) was introduced in addition to the existing sieve sizes, establishing the nominal maximum particle size for SMA-7 mixtures at 6.35 mm. Given the relatively small particle sizes of the SMA-7 mixture, the traditional classification boundaries for coarse and fine aggregates in asphalt mixtures were found to be inadequate. Therefore, it was necessary to redefine the boundary between coarse and fine aggregates. According to the Beley method, the boundary size between coarse and fine aggregates is determined as 0.22 times the nominal maximum particle size, resulting in a boundary size of 1.18 mm.

**2.3.2 Coarse aggregate gradation design.** Coarse aggregates play a crucial role in forming the skeletal structure of asphalt mixtures, which significantly impacts the performance of asphalt mixtures. To optimize the coarse aggregate gradation for SMA-7 mixtures, two methodologies are employed: multi-point supporting skeleton gradation theory and gradual filling gradation optimization theory. The first method is used to theoretically determine the optimal blending ratio for coarse aggregates of different particle sizes, while the second method employs experimental procedures to optimize the actual aggregate mix, ensuring the effective formation of the skeleton structure. The design process follows these steps:

(1) Multi-point supporting skeleton gradation:

The density of each aggregate size is first determined, followed by the calculation of the volume filling rate for each grade of coarse aggregate using equation (1). This allows for precise control over the proportion of different-sized aggregates in the mixture, thereby optimizing the overall gradation.

$$v_1 = \frac{v_0}{(1 + d/D)^3}$$
(1)

where $v_0$ is the volume filling rate of coarse aggregates when filled individually (%); $v_1$ is the volume percentage of coarse aggregates in the V-S model; D is the particle size of coarse aggregates (mm); and d is the particle size of fine aggregates (mm).

(2) Gradual filling gradation optimization:

This method optimizes the blending ratios of coarse aggregates of different particle sizes through packing density and void in coarse aggregate (VCA) tests. The process begins by determining the optimal ratio between 6.35–9.5 mm and 4.75–6.35 mm aggregates, which is then fixed. Subsequently, the optimal ratios for 2.36–4.75 mm and 1.18–2.36 mm aggregates are determined in turn.

In this approach, aggregates of varying sizes (0%, 10%, 20%, 30%, 40%, 50%, 60%, 70%, 80%, 90%, 100%) are added to the larger aggregates, followed by packing density and VCA testing. The optimal ratios, based on the best performance in the tests, are selected to determine the filling proportion between the current aggregate and the larger aggregate. Therefore, final optimal gradation for each particle size is established.

**2.3.3 Fine aggregate gradation design.** The fine aggregate gradation for SMA-7 mixtures was optimized using the n-method, with packing density and VCA as evaluation criteria. Studies have shown that finer aggregates improve high-temperature performance. The gradation curve becomes more parabolic when the n-value is between 0.3 and 0.7, leading to better compaction. However, reducing the n-value from 0.3 can also enhance aggregate packing.

For this study, n-values were selected between 0.2 and 0.7, with intervals of 0.05, resulting in 11 gradations that were tested to identify the optimal mix.

**2.3.4 Coarse and fine aggregate proportioning using the V-S method.** The coarse-to-fine aggregate ratio in the SMA-7 mixture was determined based on the V-S design method, which primarily uses the void ratio (VV) and the voids

in mineral aggregates (VMA) as key parameters. The goal was to optimize the aggregate gradation to meet performance specifications. By applying the V-S method, the proportions of coarse and fine aggregates were determined [39,40]. The V-S design equation is as follows:

$$\begin{cases} G + g = 100 \\ \frac{g}{\rho_g} = \frac{G}{\rho_g}\left(\frac{VCA-VMA}{100}\right) \\ \frac{P_a}{\rho_g} = \left(\frac{VMA-VV}{100}\right)\frac{G}{\rho_s} \end{cases}$$

(2)

where G denotes the percentage of coarse aggregate in mineral material; g represents the percentage of fine aggregate in mineral material; $\rho_g$ stands for the density of fine aggregate mixture in g/cm³; $\rho_s$ is the density of coarse aggregate at close bulk density in g/cm³; $\rho_a$ indicates the density of asphalt in g/cm³; VCA corresponds to the Voids in Coarse Aggregates under close-packed conditions; VMA signifies the void ratio of mineral material; $P_a$ percentage of asphalt-aggregate ratio and VV represents the void volume of the asphalt mix.

**2.3.5 Gradation optimization based on gray entropy analysis.** The optimal gradation range for the SMA-7 mixture has been established in the previous steps. This study further introduces gray entropy analysis to identify the key sieve sizes that significantly influence the mixture's high-temperature performance. The process begins by normalizing the data, bringing all parameters into a comparable range to allow for meaningful comparisons.

With VCA and packing density as the target properties, the gray correlation coefficient was then calculated between a reference sequence $X_0$ and a comparison sequence $X_i$ using the following formula:

$$\xi(X_0, X_i) = \frac{\min\left|X_0(k) - X_i(k)\right|}{\max\left|X_0(k) - X_i(k)\right|}$$

(3)

where ξ represents the correlation coefficient and k is the index of data points. This coefficient measures the degree of similarity between the reference and comparison sequences.

Next, the gray entropy for each factor was calculated to quantify the degree of uncertainty or disorder within the data. The gray entropy of a sequence $X = (X_1, X_2, \cdots, X_m)$ is calculated as:

$$H(X) = -\sum_{i=1}^{m} X_i \log(X_i)$$

(4)

where $X_i$ represents the normalized value of the i-th factor. The gray entropy provides a measure of the complexity of each factor's behavior in relation to the others.

The sieve sizes with higher gray entropy association degrees are considered more influential in determining the asphalt mixture's performance, allowing for the identification and ranking of the most significant sieve sizes.

## 3 Results and analysis

### 3.1 SMA-7 gradation composition

The role of aggregates in forming the skeletal structure of asphalt mixtures is closely linked to the performance of SMA-7 mixtures. Therefore, determining the optimal mass ratio for each aggregate grade is crucial. Table 5 shows the optimal proportions of coarse and fine aggregates based on the gradual filling method.

To identify the key sieve sizes, based on the optimal gradation ranges, the mass fractions of different particle sizes were adjusted. The packing density and VCA were analyzed through gray correlation entropy. Given the similar proportions across four aggregates, their influence was not considered here, and the results are shown in Table 6.

This indicates that 0.60 mm, 1.18 mm, and 4.75 mm are the key sieves for SMA-7, as the proportion of these particle sizes is critical for determining the asphalt mixture's performance.

The key sieve sizes and internal proportions for coarse and fine aggregates have been determined. For a specific mineral distribution, the V-S method (Equation 2) was used to calculate the optimal proportions. Referencing standard SMA requirements, with VMA set at 18% and VV at 4%, and using the measured densities, the coarse and fine aggregate proportions were calculated, as shown in Table 7.

The results indicate that the coarse-to-fine aggregate ratio for SMA-7 is approximately 75:25. As the grade of CB increases, the recommended proportion of coarse aggregate slightly increases, along with the asphalt-aggregate ratio ($P_a$). Due to the higher porosity and water absorption of calcined bauxite, the asphalt content is increased to meet the required performance specifications. The final sieve analysis results for the asphalt mixtures are shown in Table 8, and the optimal gradation curves for different aggregates of SMA-7 are shown in Fig 5.

## 3.2 Pavement performance evaluation

The high-temperature rutting resistance, low-temperature crack resistance, and water stability of SMA-7 asphalt mixtures with different aggregates were evaluated using the high-temperature rutting test, low-temperature bending test, freeze-thaw splitting, and immersion Marshall stability test. The results are shown in Table 9.

Table 5. Optimal proportions of coarse and fine aggregates based on stepwise filling theory.

| Type | Percentage for coarse aggregate (%)[1] | | | | Percentage for fine aggregate (%)[2] | | | | |
|---|---|---|---|---|---|---|---|---|---|
| Size(mm) | 6.35-9.5 | 4.75-6.35 | 2.36-4.75 | 1.18-2.36 | 0.60-1.18 | 0.03-0.60 | 0.15-0.30 | 0.075-0.15 | <0.075 |
| 88#CB | 10.3 | 36.5 | 26.8 | 26.4 | 15.5 | 13.4 | 11.3 | 9.5 | 50.2 |
| 85#CB | 9.8 | 35.0 | 28.7 | 26.5 | 15.2 | 13.6 | 10.8 | 9.4 | 51.0 |
| 75#CB | 8.9 | 32.4 | 30.4 | 28.3 | 14.7 | 13.2 | 11.2 | 10.1 | 50.8 |
| LS | 8.6 | 33.0 | 30.3 | 28.1 | 15.1 | 12.4 | 11.8 | 10.2 | 50.5 |

[1] Calculated based on stepwise filling method; [2] n-value of 0.25 is the optimal result, corresponding to the proportions above.

Table 6. Gray correlation results.

| | Entropy relation of coarse aggregate | | | | Entropy relation of fine aggregate | | | |
|---|---|---|---|---|---|---|---|---|
| Size(mm) | 6.35-9.5 | 4.75-6.35 | 2.36-4.75 | 1.18-2.36 | 0.60-1.18 | 0.03-0.60 | 0.15-0.30 | 0.075-0.15 |
| Packing density | 0.973 | 0.975 | 0.974 | 0.975 | 0.974 | 0.972 | 0.967 | 0.971 |
| VCA | 0.974 | 0.975 | 0.973 | 0.974 | 0.974 | 0.972 | 0.970 | 0.972 |

Table 7. Determination of coarse and fine aggregate proportions based on V-S design method.

| Type | $\rho_g$ (g/cm³) | $\rho_s$ (g/cm³) | VCA | $\rho_a$ (g/cm³) | G | g | $P_a$ |
|---|---|---|---|---|---|---|---|
| 88#CB | 2.761 | 1.743 | 32.1% | 1.024 | 77.7% | 22.3% | 6.41% |
| 85#CB | 2.717 | 1.762 | 33.5% | | 76.1% | 23.9% | 6.22% |
| 75#CB | 2.578 | 1.714 | 34.1% | | 75.4% | 24.6% | 6.33% |
| LS | 2.546 | 1.689 | 36.4% | | 72.3% | 27.7% | 6.16% |

**Table 8. Sieve passing rates for asphalt mixtures with different aggregates.**

| Sieve Size (mm) | 9.5 | 6.35 | 4.75 | 2.36 | 1.18 | 0.6 | 0.3 | 0.15 | 0.075 |
|---|---|---|---|---|---|---|---|---|---|
| 88#CB | 100 | 91.97 | 63.61 | 42.79 | 22.28 | 18.82 | 15.83 | 13.31 | 11.19 |
| 85#CB | 100 | 92.54 | 65.91 | 44.07 | 23.90 | 20.27 | 17.02 | 14.44 | 12.19 |
| 75#CB | 100 | 93.29 | 68.86 | 45.94 | 24.60 | 20.98 | 17.74 | 14.98 | 12.50 |
| LS | 100 | 93.78 | 69.92 | 48.02 | 27.70 | 23.52 | 20.08 | 16.81 | 13.99 |

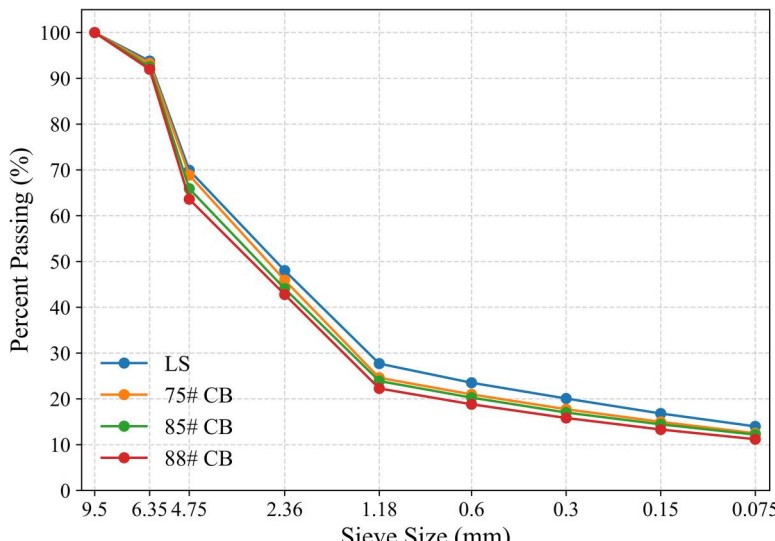

**Fig 5. Optimal gradation curves for SMA-7 mixtures with different aggregates.**

**Table 9. Performance of asphalt mixtures with different aggregates.**

| Type | Dynamic Stability (times/mm) | Maximum Bending Strain (με) | Freeze-Thaw Splitting Residual Strength Ratio (%) | Immersion Residual Stability (%) |
|---|---|---|---|---|
| 88#CB | 8847 | 3482 | 94.8 | 93.8 |
| 85#CB | 8226 | 3470 | 93.6 | 93.1 |
| 75#CB | 7038 | 3318 | 93.4 | 92.3 |
| LS | 5964 | 3306 | 96.4 | 95.4 |

Table 9 shows that the 88# CB mixture exhibits the best high-temperature stability, with a dynamic stability exceeding 8800 times/mm, approximately 1.5 times that of the LS mixture. This superior performance is due to the lower thermal conductivity of CB, which helps slow the rate of temperature rise and maintain shape stability under high temperatures. Additionally, 88# CB performs better than other aggregates in hardness, crushing value, and surface roughness, further enhancing its high-temperature performance.

For low-temperature crack resistance, the differences between the mixtures are minimal, suggesting that aggregate mechanical properties have little effect on low-temperature performance. Instead, the brittleness of the modified asphalt plays a more significant role. The higher strength of CB particles helps resist stress concentrations, reducing cracking in the asphalt matrix and improving low-temperature performance.

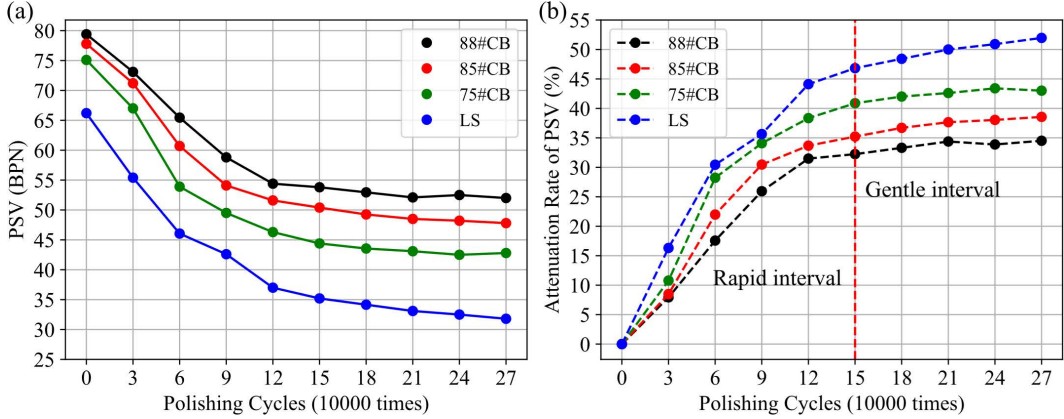

**Fig 6. PSV attenuation patterns.** (a) PSV curve; (b) PSV attenuation rate.

Water stability is primarily influenced by the adhesion between the aggregate and asphalt. As shown in Table 9, LS mixtures exhibit slightly better water stability than CB mixtures, primarily due to chemical bonding between its high CaO composition and asphalt carboxyl groups, forming hydrolysis-resistant complexes. Furthermore, as the grade of CB increases (from 75# to 88#), water absorption decreases, resulting in improved stability. The observed moisture stability hierarchy aligns with adhesion mechanisms revealed in prior study [37]. CB mixtures primarily rely on physical adhesion due to its $Al_2O_3$-dominated surface chemistry. Crucially, 88# CB exhibits enhanced water resistance approaching limestone levels (residual stability within 2%) due to reduced $SiO_2$ content and increased $Al_2O_3$ purity, which can elevating surface energy and adhesion work.

In summary, the 88# CB mixture offers excellent high-temperature stability and low-temperature crack resistance, though its water stability is slightly lower than that of the limestone mixture.

### 3.3 Aggregate polishing and PSV attenuation analysis

The PSV of aggregates plays a critical role in assessing their frictional performance, particularly in understanding the friction attenuation mechanisms under road polishing conditions. Fig 6 illustrates the PSV attenuation behavior and the PSV attenuation rate of four types of aggregates across various polishing cycles. The PSV attenuation rate is defined as the percentage change in PSV from the initial value, calculated based on the difference between the PSV at each polishing cycle and the initial PSV. This attenuation rate serves as a key indicator of the aggregate's long-term resistance to polishing, reflecting its durability under road use.

The data indicate that as the polishing cycles progress, the PSV of all aggregates declines markedly, which suggests a smoothening of the aggregate surface. Notably, the PSV attenuation of the four aggregate types exhibits a characteristic two-phase behavior: an initial rapid decay followed by a gentle stabilization phase. Based on the attenuation rate shown in Fig 6b, it can be inferred that the aggregates transition to a relatively stable plateau after approximately 120,000–180,000 polishing cycles. This plateau period marks the phase where the PSV no longer experiences significant decay. Consequently, for the purpose of this study, the first 150,000 polishing cycles are identified as the rapid decay phase for all aggregate types.

To further quantify the decay process, drawing from common methods in road engineering for characterizing long-term aggregate performance, three regression-based mode—exponential, logarithmic, and power-law—are evaluated for their ability to represent PSV attenuation, with the power-law model showing the best fit and trend, as illustrated in Fig 7. Table 10 presents the regression results for these four aggregate types, including parameters and their significance. The power-law function is described in equation (5):

$$PSV = A + W \cdot \exp^{-CN} \tag{5}$$

where, A represents the baseline PSV indicating initial polishing resistance; W is a weight factor that reflects the range of PSV variation; C represents the decay rate, which is the decisive factor for the decay speed; N is the number of polishing cycles. These parameters can be used as a reference for the performance of polishing resistance.

The regression results (Table 10) and visual trends (Fig 6) jointly confirm the superiority of CB over LS aggregates in both initial skid resistance and long-term anti-wearing durability. Specifically, CB exhibits significantly higher baseline PSV (parameter A), attributable to its corundum-phase crystals ($\alpha$-$Al_2O_3$) with Moh's hardness ≥9.0 (Table 4), which resist

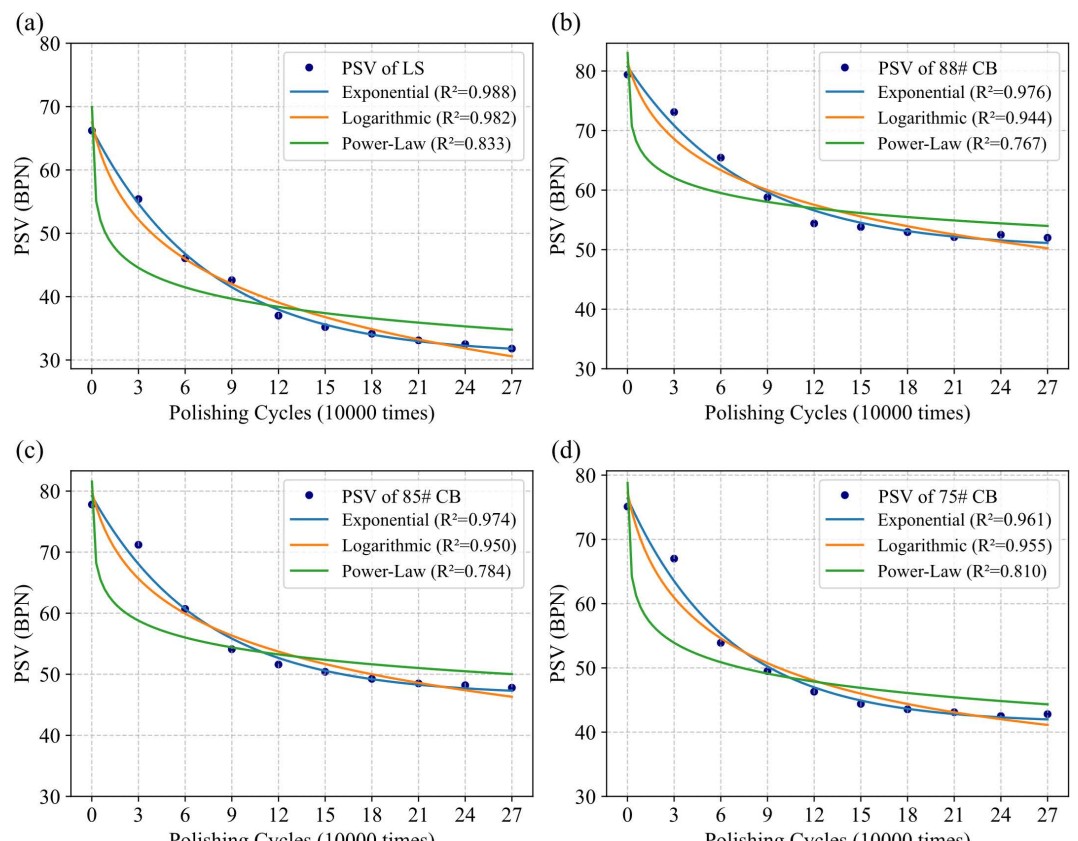

**Fig 7. The regression performance for different models in several aggregates.** (a) LS; (b) 88#CB; (c) 85# CB; (d) 75# CB.

**Table 10. Regression analysis for PSV attenuation with polishing cycles.**

| Type | Parameters | | | R² |
|------|------------|---|---|----|
| | **A** | **W** | **C** | |
| **88#CB** | 50.226 | 30.511 | 0.131 | 0.976 |
| **85#CB** | 46.543 | 32.654 | 0.139 | 0.974 |
| **75#CB** | 41.425 | 33.980 | 0.143 | 0.961 |
| **LS** | 35.825 | 35.651 | 0.138 | 0.988 |

plastic deformation under tire polishing. Furthermore, CB achieves lower attenuation amplitude (parameter W), reflecting its dense sintering-derived texture that inhibits crack propagation and aggregate spalling.

Notably, the 88#CB aggregates exemplify optimal performance, demonstrating the lowest attenuation range (W = 30.511) and slowest decay rate (C = 0.131), enabling its PSV to stabilize after 120,000 polishing cycles. In contrast, LS aggregates show the weakest long-term resistance to polishing, with the lowest initial PSV (A = 35.825) and highest attenuation amplitude (W = 35.651). Their calcite-dominated composition lacks intrinsic resistance to plastic flow, causing PSV to drop below 50% of its initial value and continues to degrade even after 150,000 polishing cycles.

Moreover, parameter C explicitly governs the initial attenuation kinetics of PSV curves, as evidenced by its significant variation across aggregates and distinct profiles in Fig 7. For instance, the 75#CB aggregate shows the fastest initial decay (C = 0.143), which results in a sharper decline in PSV during the early polishing stages. Conversely, LS aggregates display a moderate C value (0.138) despite higher long-term degradation, owing to their prolonged attenuation period (>180,000 cycles) which reduces the initial curve slope.

The analysis also suggests that the PSV formulation of Equation 5 can serve as a basis for selecting anti-skid aggregates. The optimal aggregates should possess superior initial strength, a lower attenuation rate, and a smaller attenuation magnitude, which correspond to higher A values and lower W and C values, respectively.

### 3.4 Mixture skid resistance performance analysis

**3.4.1 MPD attenuation and surface topography.** Ensuring sufficient and durable skid resistance is crucial for ultra-thin wear courses. Although the PSV distribution of aggregates provides an indirect indication of wear resistance, asphalt mixture testing is essential for validation. Table 11 presents the MPD test results for SMA-7 asphalt mixtures prepared with different types of aggregates before and after polishing.

As shown in Table 11, all mixtures initially exhibited similar surface textures due to the consistent SMA-7 gradation. However, after 50,000 polishing cycles, the MPD of all mixtures decreased by approximately 13–22%. This reduction indicates significant degradation of both macro and micro textures, potentially leading to a decline in skid resistance.

Among the tested aggregates, those with higher hardness and wear resistance, such as 88#CB and 85#CB, exhibited smaller MPD reductions, thereby maintaining deeper surface textures. This is attributed to their resistance to abrasion, which supports the skeleton structure of the asphalt mixture and minimizes surface texture loss. Additionally, the higher proportion of coarse aggregates in CB-SMA-7 mixtures (as shown in Table 7) further enhances MPD retention, helping to stabilize the surface properties of the mixture.

To characterize micro-textural evolution post-polishing, Fig 8 depicts surface topography within wheel-track zones after 50,000 cycles. All mixtures showed significant texture degradation manifested through detached bitumen films, exposed aggregates, and overall smoothing from bitumen infill between particles. These morphological shifts directly explain observed MPD reductions, such as LS decreasing from 0.524 mm to 0.402 mm.

Critical differences emerged in aggregate durability, where LS mixtures suffered the most severe abrasion. Aggregates are polished nearly smooth and only sporadic edges protruding (Fig 8a). This accelerated abrasion aligns with the inherent softness of limestone minerals, where shear forces readily smooth exposed surfaces. Conversely, 88#CB mixtures retained pronounced aggregate relief with distinct protrusions (Fig 8d), owing to their engineered mineralogy resisting deformation.

**Table 11. MPD test results.**

| Polishing cycles (times) | LS | 75#CB | 85#CB | 88#CB |
|---|---|---|---|---|
| 0 | 0.524 | 0.529 | 0.537 | 0.541 |
| 50000 | 0.402 | 0.426 | 0.448 | 0.462 |

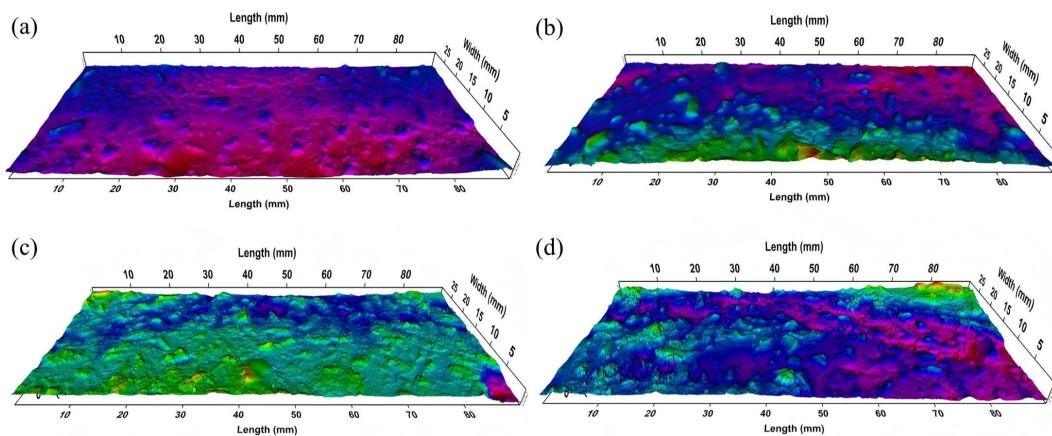

**Fig 8. Surface texture morphology of pavements.** (a) LS; (b) 75#CB; (c) 85#CB; (d) 88#CB.

It can be inferred that the sustained surface roughness in 88#CB maintains friction through direct mechanical interlock after prolonged polishing. Meanwhile, LS's smoothed surface compromises this mechanism. Additionally, the higher retained PSV of CB aggregate enhances frictional resistance at tire-pavement interfaces, synergistically boosting overall skid resistance.

**3.4.2 Dynamic friction coefficient variation under different polishing times.** Fig 9 shows the variation of the dynamic friction coefficient (DF) of SMA-7 wearing courses with polishing cycles at simulated speeds of 20, 40, 60, and 80 km/h. Overall, the DF of the mixture specimens decreased significantly with increasing polishing cycles. After 20,000 cycles, the rate of attenuation slowed, entering a stable plateau phase, which aligns with the PSV curve. Furthermore, across all polishing cycles and speed levels, CB mixtures consistently exhibited higher DF values compared to LS mixtures, particularly in terms of attenuation rate and attenuation ratio. Detailed data are provided in Table 12. The attenuation rate refers to the average reduction in DF per 10,000 polishing cycles, while the attenuation ratio indicates the percentage decrease in DF relative to the initial value after polishing.

After 50,000 polishing cycles, the average attenuation ratios of the DF for 88#CB, 85#CB, and 75#CB mixtures were 33.7%, 37.6%, and 41.2%, respectively, whereas LS exhibited an attenuation ratio of 46.3%. For newly prepared specimens, the difference in DF value of four mixtures was less than 0.1. However, after extensive polishing (>30,000 cycles), the average DF for LS dropped to approximately 0.34, which is only 57.2% of that for 88#CB and below the critical lateral force coefficient value of 0.35 used in roadway pavement design.

Regarding the attenuation rate, 88#CB mixtures demonstrated a significantly lower friction attenuation rate compared to other mixtures. As shown in the trends in Fig 9, both 88#CB and 85#CB mixtures required more than 30,000 polishing cycles to reach the plateau phase, maintaining relatively high friction resistance. In contrast, LS mixtures typically reached the plateau phase after only 10,000–20,000 cycles. This indicates that high-grade CB aggregates exhibit a longer wear resistance cycle than traditional aggregates, which could help reduce the maintenance frequency and duration of anti-slip thin layers in practical applications.

The simulated speed also had a significant impact on the dynamic friction coefficient. For 50,000 polishing cycles, as speed increased from 20 km/h to 80 km/h, the DF decreased by 17.8% on average. This attenuation was non-linear: the friction decay rate at 20–40 km/h was 36% higher than at 60–80 km/h. Such speed-dependent behavior originates from the transition in friction dominance.

In the low-speed regime (≤40 km/h), micro-texture governs friction primarily, where high contact frequency enhances shear stress transmission and maximizes the effectiveness of PSV-driven adhesive friction as the dominant contributor

[41]. Conversely, in the high-speed regime (≥60 km/h), reduced tire-road contact time diminishes the influence of micro-texture, and macro-texture dominates by inducing hysteretic energy dissipation through bulk rubber deformation. Although the absolute friction force is comparatively lower at high speeds, the relative attenuation of DF weakens due to the limited decay of MPD across aggregates.

For the lower speed groups (Fig 9a and 9b), the peak friction coefficient did not occur at the initial test value but rather around 2,500 polishing cycles, reflecting a transition in friction mechanisms. Existing literature on the wear and friction behavior of asphalt pavements (Fig 10) suggests that before the aggregates begin to wear, external loads predominantly

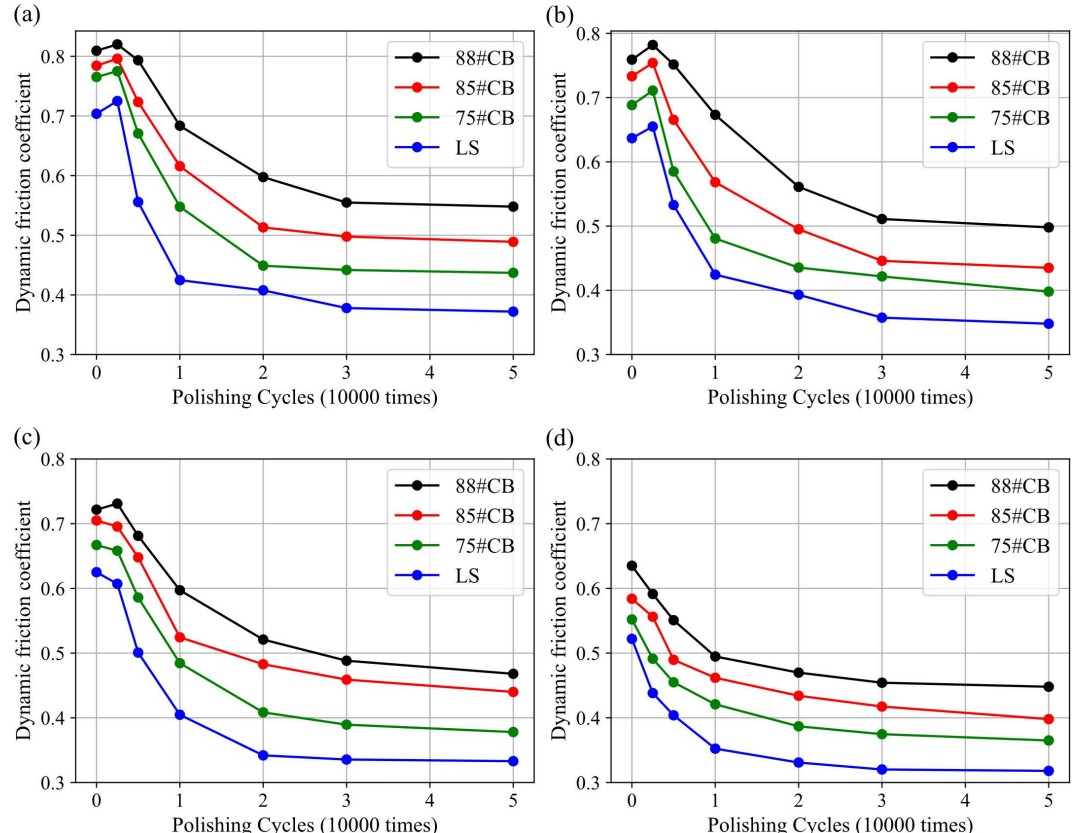

**Fig 9. Distribution of the dynamic friction coefficient under different simulated speeds.** (a) 20 km/h; (b) 40 km/h; (c) 60 km/h; (d) 80 km/h.

**Table 12. Attenuation rate and ratio of dynamic friction coefficient.**

| Velocity | Attenuation index | 88#CB | 85#CB | 75#CB | LS |
|---|---|---|---|---|---|
| 20km/h | Rate (ΔDF/$10^4$ times) | 0.054 | 0.061 | 0.068 | 0.071 |
| | Ratio (%) | 33.2 | 38.6 | 43.6 | 48.6 |
| 40km/h | Rate (ΔDF/$10^4$ times) | 0.057 | 0.063 | 0.063 | 0.064 |
| | Ratio (%) | 36.3 | 42.3 | 44.0 | 46.9 |
| 60km/h | Rate (ΔDF/$10^4$ times) | 0.053 | 0.053 | 0.057 | 0.059 |
| | Ratio (%) | 35.9 | 37.6 | 43.3 | 46.7 |
| 80km/h | Rate (ΔDF/$10^4$ times) | 0.034 | 0.036 | 0.038 | 0.042 |
| | Ratio (%) | 29.4 | 31.8 | 33.9 | 39.1 |

affect the asphalt layer on the surface [42,43]. Referring to Fig 10, during the initial 0–2,500 cycle range, asphalt film detachment exposes fresh aggregate micro-asperities, enhancing adhesive friction through increased rubber-aggregate contact area – a phenomenon termed "asphalt friction".

For polishing more than 2500 cycles, the progressive smoothing of micro-texture reduces the density of surface asperities, thereby diminishing the adhesive friction component due to reduced micro-scale contact points for rubber-aggregate interactions. Concurrently, hysteretic friction induced by macro-texture increases in relative contribution as the primary resistance mechanism. This transition explains why CB mixtures retain higher DF: high-PSV aggregates resist micro-smoothing, preserving adhesive friction capacity; higher MPD make interlocked angular particles maintain macro-voids against closure, enhancing hysteretic friction by sustaining macro-texture depth.

**3.4.3 Relationship between DF coefficient of mixtures and PSV of aggregates.** Given the strong similarity between the attenuation curves of the DF coefficient and PSV, it can be inferred that the PSV of aggregates is a key determinant in their skid resistance, especially in the texture polishing phase. Equation 6 is derived to describe the variation of the dynamic friction coefficient (DF) with respect to the polishing cycles for different mixtures.

$$DF_i = B + K \cdot \exp^{-RN} \tag{6}$$

Where $DF_i$ is the dynamic friction coefficient at the simulated speed i, B is the base friction coefficient provided by the aggregates, K is the attenuation weight factor, R reflects the rate of DF change, and N is the number of polishing cycles. To exclude the early-phase "asphalt friction" effect, we eliminate data prior to 2,500 polishing cycles. Regression analysis was performed for the representative design speeds (60 km/h and 80 km/h) of the four mixed types. The regression results are summarized in Table 13.

The regression results indicate that, among the four mixtures, the 88#CB mixture exhibits the lowest attenuation rate and weight factor K at both speed levels, suggesting the best skid resistance and stability. In contrast, LS exhibits the fastest decay rate and the lowest K, leading to significantly inferior skid resistance. The high $R^2$ values further confirm the strong correlation between the skid resistance of the mixtures and the PSV of the aggregates.

To further investigate the relationship between PSV and DF, the attenuation rates for both were calculated using their respective regression formulas, with an emphasis on comparing their correlation. These attenuation rates were derived from the first derivative of the respective functions. Due to differences in polishing and pressing methods and the significant variations in polishing cycles, distinct cycle ranges were selected: 2,500–50,000 cycles for DF and 0–150,000 cycles for PSV. The polishing cycles were divided into ten equal intervals, and the attenuation rates (derivatives) were calculated for each. The results of the correlation analysis are shown in Table 14.

The correlation analysis shows a strong positive relationship between the attenuation rates of DF and PSV for all mixtures (greater than 0.75), indicating that as the PSV attenuation rate increases, so does the DF attenuation rate. Moreover, the correlation is stronger at lower speeds, which implies that at low speeds, skid resistance is more closely related to the aggregate's properties. These speed-dependent interactions align with Persson's friction theory [44,45].

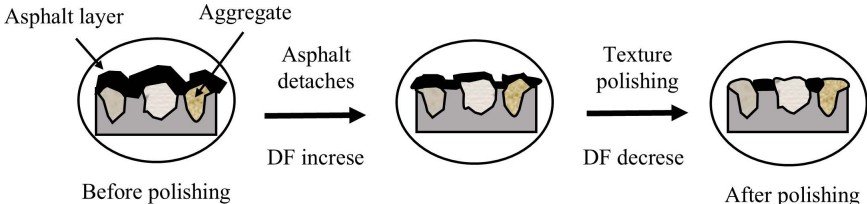

**Fig 10. Texture polishing process of asphalt pavement.**

**Table 13. Regression analysis for DF attenuation with polishing cycles.**

| DF | Type | Parameters | | | $R^2$ |
|---|---|---|---|---|---|
| | | B | K | R | |
| $DF_{60}$ | 88#CB | 0.447 | 0.294 | 0.634 | 0.974 |
| | 85#CB | 0.433 | 0.300 | 0.873 | 0.964 |
| | 75#CB | 0.366 | 0.326 | 0.886 | 0.977 |
| | LS | 0.323 | 0.326 | 1.182 | 0.971 |
| $DF_{80}$ | 88#CB | 0.450 | 0.180 | 1.173 | 0.996 |
| | 85#CB | 0.407 | 0.188 | 1.269 | 0.975 |
| | 75#CB | 0.371 | 0.177 | 1.383 | 0.995 |
| | LS | 0.321 | 0.198 | 1.860 | 0.997 |

**Table 14. Correlation matrix between attenuation rate of DF and PSV.**

| $R^2$ | PSV | $DF_{20}$ | $DF_{40}$ | $DF_{60}$ | $DF_{80}$ |
|---|---|---|---|---|---|
| PSV | 1 | | | | |
| $DF_{20}$ | 0.934 | 1 | | | |
| $DF_{40}$ | 0.902 | 0.977 | 1 | | |
| $DF_{60}$ | 0.873 | 0.924 | 0.984 | 1 | |
| $DF_{80}$ | 0.858 | 0.893 | 0.936 | 0.922 | 1 |

High PSV aggregates resist asperity blunting, preserving micro-scale contact points. This directly sustains adhesive friction through enhanced rubber-asperity interactions, explaining the highest 0.934 correlation between PSV decay and DF decay at 20 km/h; while at higher speeds, macro surface texture controls hysteresis friction, which plays a more significant role in determining skid resistance [46]. Even high-PSV mixtures (88#CB) show DF decay if macro-voids collapse (MPD < 0.470 mm). Thus, PSV-DF correlation drops to 0.61 at 80 km/h.

From the static perspective, Fig 11 illustrates the numerical relationship between PSV and DF before and after polishing. The results show that, after prolonged polishing (Fig 11b), a clear linear relationship emerges between PSV and DF, with a higher slope observed at lower speeds, quantifying the heightened influence of micro-texture on adhesive friction under low-speed conditions. In contrast, unpolished mixtures (Fig 11a) exhibit a near-flat PSV-DF curve, especially for PSV below 77.5, attributable to asphalt film coverage masking aggregate micro-texture. Polishing activates PSV's role by exposing asperities, enabling adhesive friction via micro-scale contact.

As PSV intrinsically reflects aggregate's micro-scale skid resistance and wear resistance, these findings suggest that the aggregate's inherent wear resistance directly governs the long-term skid performance of asphalt mixtures, underscoring the critical role of high-quality aggregates in ensuring pavement safety durability.

## 4 Discussion

### 4.1 Skid resistance mechanisms of CB aggregates

The superior skid resistance performance of CB mixtures, as demonstrated in the results, can be attributed to a dual mechanism: the preservation of surface texture and the inherent high skid resistance of the aggregates. At the microscale, the analysis of aggregate properties reveals that high-hardness aggregates such as 88#CB play a key role in maintaining surface texture. As shown in Table 4, the primary crystalline phases of CB aggregates include corundum and mullite, both of which have higher Mohs hardness values compared to LS aggregates. Fundamentally, the progressive increase in $Al_2O_3$ content from 75#CB (75.53%) to 85#CB (85.24%) and 88#CB (90.29%) (Table 3) elevates the corundum

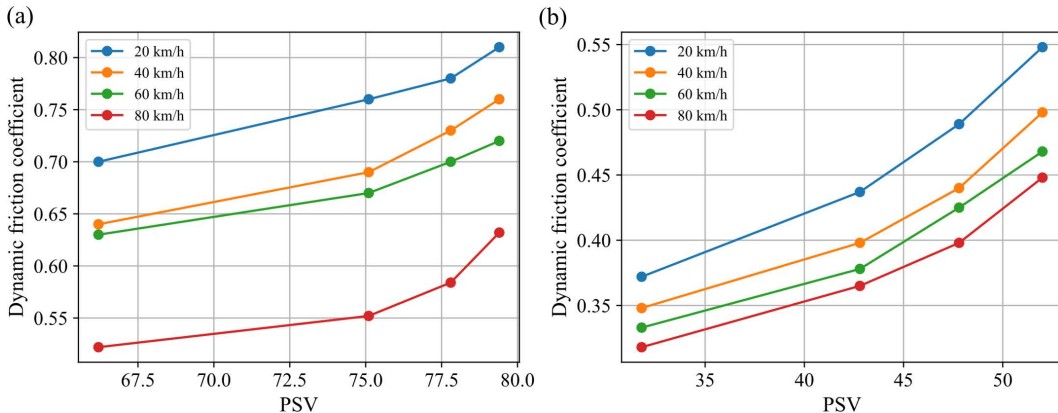

**Fig 11. Relationship between PSV and DF.** (a) Before polishing; (b) After polishing.

phase proportion (76.0%→84.0%→87.0%) while reducing softer mullite (Table 4). This mineralogical shift increases the Aggregate Hardness Parameter (9.78→9.89→10.08), providing superior resistance to plastic deformation during polishing. The significantly higher hardness of these minerals contributes to superior wear resistance, which is essential for maintaining long-term surface texture. Moreover, the Mohs hardness ratio of corundum and mullite is 9:6. During polishing cycles, the "soft" mullite phase is preferentially worn, while the "hard" corundum phase forms micro-protrusions on the aggregate surface. According to recent literature, this dual-phase structure creates a self-renewing micro-texture that delays the flattening of asperities [28]. This mechanism is empirically evidenced by the higher initial PSV, slower decay rates (C parameter in Table 10), lower PSV attenuation range (W parameter), and earlier PSV stabilization observed in higher-grade CB aggregates.

At the macro-scale, the optimized gradation with a higher proportion of coarse aggregates, particularly the robust 88#CB particles, enhances the asphalt mixture's texture by increasing the overall coarse aggregate fraction by 5.4% compared to LS (Table 7). This leads to a more stable interlocking structure, contributing to a deeper and more durable surface texture. The inherent high resistance to crushing and enhanced angularity/texture of CB aggregates (Fig 8), further fortifies this skeleton against deformation. The high resistance to wear allows 88#CB mixtures to sustain higher dynamic friction coefficients over extended polishing cycles. This structural advantage is reflected in the 35.2% lower MPD attenuation value for 88#CB compared to LS after 50,000 polishing cycles. Additionally, the lower thermal conductivity of CB aggregates helps mitigate rapid temperature rise within the mixture under high ambient temperatures, synergizing with the skeletal structure to maintain surface texture under heavy traffic conditions (dynamic stability 1.5 times higher than LS).

Furthermore, PSV plays a crucial role in determining the long-term skid resistance of aggregates. As the asphalt binder wears off over time, aggregates come into direct contact with vehicle tires, with PSV serving as a key indicator for maintaining friction. 88#CB aggregates exhibit a lower PSV attenuation rate compared to LS, allowing them to maintain higher frictional properties for a longer period. This slower attenuation, driven by the mineral-hardness hierarchy established earlier, not only ensures that 88#CB retains its strength longer but also extends its effective lifespan under traffic-induced wear. Research supports this, indicating that aggregates with a higher PSV are more resistant to wear and maintain their frictional properties for longer durations, thus enhancing the skid resistance of the asphalt mixture, even after extensive polishing cycles [34,47].

In addition to PSV, the adhesion between the aggregate and the asphalt binder is critical for maintaining friction. CB aggregates provide a stable bond due to their high angularity and excellent adhesion properties. The observed performance gradient among CB aggregates, particularly the marked superiority of 88#CB over both 85#CB and 75#CB, carries

significant practical implications for pavement engineering. Its slower PSV decay rate and higher residual DF coefficient directly enhance long-term safety on critical sections (e.g., curves, ramps) where low-speed friction dominates. The extended period required for the friction coefficient of 88#CB mixtures to reach a plateau phase (>30,000 cycles compared to ~10,000–20,000 for LS) indicates a potentially longer functional service life for the ultra-thin wearing course before reaching critical skid resistance thresholds, reducing maintenance frequency and associated traffic disruptions over the pavement lifecycle.

### 4.2 Potential strategies for improving skid resistance

The findings advocate for high-hardness CB aggregates in skid-critical applications. Notably, the selection between specific CB grades, particularly 88#CB and the more common 85#CB, necessitates lifecycle cost-benefit analysis. While 88#CB delivers optimal long-term performance, its higher $Al_2O_3$ content and purity (Table 3) typically incurs production cost premiums (for about 10%~20%) compared to 85#CB. Therefore, a pragmatic approach is recommended: for pavement sections where maximizing safety and extending the maintenance-free period are of utmost priority, such as high-risk locations or heavily trafficked corridors, the investment in 88#CB aggregates is strongly justified by its superior long-term performance. In scenarios where initial cost constraints are tighter, or for roadways with moderate traffic volumes, 85#CB still offers a substantial performance. Crucially, both 88#CB and 85#CB provide far superior performance compared to limestone, making them compelling choices when lifecycle costs and safety are key drivers.

The inverse correlation between vehicle speed and dynamic friction coefficient (DF) observed in this study highlights the need for speed-specific design strategies. At lower speeds, aggregates with PSV enhance molecular adhesion between the tires and the pavement surface, improving friction. However, at higher speeds, macrotexture-dominated mechanisms become more influential [46,48]. Therefore, optimized gradation, such as using SMA-5 or SMA-7, is required to maintain sufficient MPD and mitigate the risks of hydroplaning. In road sections of sharp curves or small-radius ramps, it is especially beneficial to prioritize high-PSV aggregates to resist the loss of micro texture caused by polishing. These strategies align with established theories of texture-wavelength interaction, ensuring that the pavement's surface remains effective under various traffic conditions [49].

The inflection point at 30,000 polishing cycles (Fig 9) signifies a transition phase where mixtures shift from rapid DF decay (0.054–0.057/10⁴ times) to stabilized skid resistance (<0.02/10⁴ times). This implies that preventive maintenance prior to 30,000 cycles could preserve 85–90% of initial texture depth, avoiding irreversible skeleton collapse caused by aggregate reorientation. However, direct extrapolation to field conditions requires caution – actual traffic polishing involves complex interactions of axle loads, tire types, and environmental aging not captured in accelerated tests. To bridge this gap, calibration factors derived from field-core polishing tests can convert laboratory cycles to equivalent annual average daily traffic (AADT). Agencies should further define site-specific critical thresholds (e.g., MPD < 0.45mm & DF80 < 0.45 for 88#CB) through embedded texture sensors, enabling data-driven maintenance scheduling that balances safety and economy.

### 4.3 Limitations and future work

This study provides important insights into the role of aggregate properties, specifically PSV and hardness, in determining the skid resistance of asphalt mixtures. However, several limitations must be acknowledged. First, the study did not explore the impact of gradation variations or the aggregate-binder ratios on the skid resistance performance of the mixtures. Future studies could consider varying the gradation of mixtures and the asphalt content to assess how these factors influence the long-term performance of the asphalt mixtures.

Additionally, only LS aggregates were used as a control in this study. Including other types of aggregates, such as basalt or granite, could provide a more comprehensive understanding of the relationship between aggregate properties and skid resistance. Moreover, while laboratory-based tests provide valuable insights, further research using field testing

and real-world conditions could offer a more accurate representation of how these mixtures perform under traffic loads, environmental changes, and over extended periods.

Notably, this study mainly focused on aggregate properties' impact on long-term skid resistance, yet the results indicate that LS's initial moisture resistance (95.4% residual stability) is slightly higher than 88#CB (93.8%), its long-term skid performance under environmental aging may exhibit compensatory advantages. Future work should quantify texture retention mechanisms during freeze-thaw cycles, particularly how CB's alumina-dominated microstructure mitigates friction loss when chemical bonds degrade. Field validation across climatic zones is also warranted to establish predictive models for skid life extension.

The study also focused on the PSV of aggregates and its effect on frictional performance. Further research could examine other characteristics of aggregates, such as angularity, shape, and texture, and their combined effects on the long-term durability of road surfaces. Additionally, exploring alternative surface treatments or additives that could enhance the wear resistance of aggregates might offer new approaches to improving the skid resistance of asphalt pavements.

## 5 Conclusions

This study systematically examined the long-term polishing behavior of ultra-thin friction courses using three grades of CB aggregates and conventional LS. The research focused on optimizing the SMA-7 gradation and analyzing the relationship between the aggregates' polishing resistance and the skid resistance of the asphalt mixtures. The findings highlight the significant influence of aggregate quality on pavement performance, particularly in terms of long-term wear resistance and frictional behavior. Key findings from the study include:

(1)  The maximum particle size of SMA-7 was determined to be 6.35 mm, with a recommended ratio of coarse-to-fine aggregate ratio of 75:25. The higher porosity and angularity of CB aggregates necessitate a marginally increased asphalt content to ensure effective skeleton interlock while maintaining VMA compliance. This balanced design enables uniform texture depth across mixtures.

(2) The 88#CB-SMA-7 mixture exhibited superior high-temperature stability, achieving a dynamic stability of 8,847 cycles/mm—1.5 times that of LS (5,964 cycles/mm). This superior rutting resistance stems from CB's lower thermal conductivity and robust aggregate structure that maintains shape stability under thermal loading.

(3) CB aggregates exhibited slower PSV decay, with 88#CB stabilizing after 120,000–150,000 cycles, achieving a 30% reduction in attenuation rate compared to LS. This performance hierarchy correlates with mineral hardness metrics, confirming corundum-phase dominance governs long-term wear resistance.

(4) CB mixtures maintained higher dynamic friction coefficients throughout polishing, particularly for 88#CB, which shows 29.4–36.3% greater residual friction than LS after 50,000 cycles. The friction decay inflection point at 30,000 cycles reveals CB's extended service plateau, contrasting LS's premature stabilization within 10,000–20,000 cycles.

(5) Strong correlations ($R^2 > 0.85$) between PSV decay and friction coefficient attenuation, especially at lower speeds (≤40 km/h), verify that aggregate-scale polishing resistance fundamentally controls pavement-scale skid performance. CB's stable micro-texture preserves friction-generating asperities under prolonged polishing.

These findings underscore the critical role of aggregate quality in enhancing long-term pavement performance, particularly in critical sections such as curves, ramps, and approaches where low-speed friction is paramount. The superior wear resistance and frictional properties of CB aggregates, particularly 88#CB, offer significant operational benefits by potentially reducing maintenance frequency and associated traffic disruptions over the pavement's lifecycle.

## Supporting information

**S1. Data.** The experimental data obtained through the DF test and the PSV attenuation test.
(CSV)

(2) Gradual filling gradation optimization:

## Acknowledgments

The author would like to thank roadway pavement laboratory of CCCC Second Highway Engineering Bureau Third Engineering Company for their support in this study.

## Author contributions

**Conceptualization:** Pengfei Li, Lingkun Kong.

**Data curation:** Lingkun Kong, Chenwei Gu.

**Formal analysis:** Pengfei Li, Chenwei Gu.

**Funding acquisition:** Chenwei Gu.

**Investigation:** Pengfei Li, Nan Mao.

**Methodology:** Pengfei Li, Nan Mao.

**Project administration:** Chenwei Gu.

**Resources:** Pengfei Li, Lingkun Kong.

**Software:** Lingkun Kong, Nan Mao.

**Supervision:** Nan Mao, Chenwei Gu.

**Validation:** Lingkun Kong, Nan Mao, Chenwei Gu.

**Visualization:** Chenwei Gu.

**Writing – original draft:** Pengfei Li, Lingkun Kong.

**Writing – review & editing:** Pengfei Li, Nan Mao, Chenwei Gu.

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
