## [Decision Letter · Decision Letter 0]

Dear Dr. Gu,

Thank you for submitting your manuscript to PLOS ONE. After careful consideration, we feel that it has merit but does not fully meet PLOS ONE’s publication criteria as it currently stands. Therefore, we invite you to submit a revised version of the manuscript that addresses the points raised during the review process.

We look forward to receiving your revised manuscript.

Kind regards,

Jiaolong Ren

Academic Editor

PLOS ONE

Reviewers' comments:

Reviewer's Responses to Questions

**Comments to the Author**

1. Is the manuscript technically sound, and do the data support the conclusions?

Reviewer #1: Yes

Reviewer #2: Yes

2. Has the statistical analysis been performed appropriately and rigorously?

Reviewer #1: N/A

Reviewer #2: Yes

3. Have the authors made all data underlying the findings in their manuscript fully available?

Reviewer #1: Yes

Reviewer #2: Yes

4. Is the manuscript presented in an intelligible fashion and written in standard English?

Reviewer #1: Yes

Reviewer #2: Yes

Reviewer #1: 1- The regression model used to describe the Polished Stone Value attenuation with polishing cycles (Equation 5) is well fitted with high R² values. However, the physical interpretation of parameters A, W, and especially the decay rate C requires deeper discussion. The paper should elaborate on the sensitivity of these parameters to aggregate mineralogy and texture, and whether these parameters can be generalized beyond the tested aggregates. Additionally, the choice of an exponential decay model should be justified with comparison to other possible models, such as logarithmic or power law, which are commonly used in wear and polishing studies.

2- The DF attenuation results at varying speeds reveal an expected decrease in DF with increasing speed. However, the experimental design would benefit from including a more detailed discussion on the effect of speed-dependent contact mechanics, specifically how microtexture and macrotexture interactions change with speed and how this correlates with real-world tire-pavement friction. The paper currently states that low speeds show stronger correlation with PSV but could further integrate tribological principles to explain the observed trends in greater depth.

3- The results for MPD attenuation indicate that CB aggregates retain surface texture better than limestone. Yet, the study should clarify the relationship between MPD and skid resistance more explicitly, considering that MPD alone may not capture microtexture variations which critically affect friction. A combined analysis involving both macrotexture and microtexture measurements, possibly using advanced surface profilometry, would strengthen the interpretation of texture stability.

4- While the paper reports superior performance of 88# CB over 75# and 85# CB, it does not sufficiently explain the practical significance of these differences in field conditions. The authors should discuss whether the incremental improvement from 85# to 88# CB justifies potential cost or availability trade-offs. Moreover, the differences in chemical and physical properties between these grades could be linked more explicitly to their observed performance metrics to clarify material selection criteria.

5- The study finds that limestone mixtures have marginally better water stability than CB mixtures due to chemical bonding effects. However, the explanation could benefit from further experimental validation, such as surface energy measurements or binder-aggregate adhesion tests, to corroborate this chemical bonding hypothesis. Additionally, given that the 88# CB mixture’s water stability is close to limestone, the impact of water stability on long-term skid resistance should be discussed in the context of field aging and freeze-thaw cycles.

Reviewer #2: The manuscript titled “Impact of Calcined Bauxite Aggregates on the Polishing Resistance and Skid Resistance Performance of SMA-7 Asphalt Mixtures” addresses a pertinent research topic within the realm of structural engineering. This study investigates how using calcined bauxite as a coarse aggregate affects the surface durability and safety characteristics—specifically polishing resistance and skid resistance—of a particular type of asphalt mixture (SMA-7). The research likely aims to determine whether this aggregate can improve road safety and extend pavement life under traffic wear. While the study presents an interesting perspective, a thorough evaluation highlights several areas requiring improvement. To enhance the manuscript's clarity, it is essential to refine the articulation of key concepts, ensuring that the arguments are logically structured and effectively communicated. Additionally, the methodological approach should be strengthened by providing a more detailed explanation of the research design, data analysis techniques, and validation processes to ensure scientific rigor. Furthermore, a deeper discussion of the findings, including their broader implications and potential applications, would significantly enhance the manuscript’s impact. Addressing these aspects will improve the overall coherence, credibility, and contribution of the study, aligning it with the expected scholarly standards.

1- The introduction section should be improved. The literature review was not suitable.

2- Why was the ASTM C33 classification curve of aggregates not presented?

3- The quality and detail of the SEM images were not suitable for publication.

4- More details should be provided about PSV specimens and tests.

5- Why were no keywords not presented in the presented manuscript?

6- The quality of the provided curves is not suitable for publication.

7- The conclusion section has no depth. Improve it.

8- Lots of references are outdated. Please expand them. If a suitable position is found, authors can cite references below.

[1] Pre-and post-heating bar-concrete bond behavior of CFRP-wrapped concrete containing polymeric aggregates and steel fibers: Experimental and theoretical study. [2] Evolution of confinement stress in axially loaded concrete-filled steel tube stub columns: Study on enhancing urban building efficiency. [3] Sulfuric acid resistance of concrete containing coal waste as a partial substitute for fine and coarse aggregates. [4] The effect of sulfuric acid attack on mechanical properties of steel fiber-reinforced concrete containing waste nylon aggregates: Experiments and RSM-based optimization. [5] Flexural strengthening of heat-damaged RC beams with NSM CFRP strips and SFRC layer: Experimental evaluation and theoretical analysis.

**Do you want your identity to be public for this peer review?** For information about this choice, including consent withdrawal, please see our Privacy Policy

Reviewer #1: No

Reviewer #2: No

---

## [Author Response · Author response to Decision Letter 1]

16 Jul 2025

Dear Editors and Reviewers:

Sincerely thank you for your letter and reviewers’ comments concerning our manuscript entitled “Impact of Calcined Bauxite Aggregates on the Polishing Resistance and Skid Resistance Performance of SMA-7 Asphalt Mixtures” (PONE-D-25-25733). We appreciate the time and effort invested in the review process.

The insightful suggestions provided by the reviewers have been extremely helpful in guiding us to revise and improve the manuscript. In response to the comments, we have made substantial revisions throughout the manuscript. Specifically, the Introduction, Methods, Results, Discussion, and Conclusion sections have all been carefully revised to enhance clarity, scientific rigor, and presentation quality.

In particular, we have restructured Sections 3.3 and 3.4, refining the results and their presentation to better align with the reviewers’ feedback. Furthermore, we have updated and added several figures to improve data interpretation (adding Fig 5 Line 360, Fig 7 Line 445, Fig 8 Line 528 and update Fig 1 Line 151, Fig 2 Line171). All changes have been highlighted using the “Track Changes” feature in the revised manuscript for easy reference.

Please find below our detailed, point-by-point responses to the reviewers’ comments and an explanation of the corresponding modifications made.

We respectfully submit the revised manuscript for your consideration and look forward to the results of the editorial review.

Sincerely,

All co-authors of PONE-D-25-25733

Response to Editors

Response: The manuscript was revised and corrected based on the PLOS ONE’s style requirements.

Response: Our research was supported by the fundamental research funds for the central universities, CHD (300102213509). We will check our grant numbers in our resubmission.

Response: We will submit the data in our research by uploading the Supporting Information file.

Response to Reviewer 1

Reviewer #1:

1- The regression model used to describe the Polished Stone Value attenuation with polishing cycles (Equation 5) is well fitted with high R² values. However, the physical interpretation of parameters A, W, and especially the decay rate C requires deeper discussion. The paper should elaborate on the sensitivity of these parameters to aggregate mineralogy and texture, and whether these parameters can be generalized beyond the tested aggregates. Additionally, the choice of an exponential decay model should be justified with comparison to other possible models, such as logarithmic or power law, which are commonly used in wear and polishing studies.

Response: Thank you for your thorough and insightful comments. In response, we have compared the fitting performance of three regression models—exponential, logarithmic, and power-law in Fig. 7 to justify our selection of the power-law model (Lines 426–430).

We also added brief clarifications on the physical meaning of the model parameters: A as the baseline polishing resistance, W as the PSV variation range linked to aggregate texture and fracture behavior, and C as the decay rate governing early wear kinetics (Lines 430–446).

Additionally, we expanded our discussion to highlight how these parameters vary with aggregate mineralogy and texture—for example, how corundum-rich CB aggregates exhibit higher A and lower W, whereas calcite-dominated LS aggregates show the opposite trend (Lines 448–488).

Finally, we have noted the potential for using these parameters to guide material selection, emphasizing that the consistent physical interpretation across different aggregates supports a certain degree of generalizability (Lines 489–492).

2- The DF attenuation results at varying speeds reveal an expected decrease in DF with increasing speed. However, the experimental design would benefit from including a more detailed discussion on the effect of speed-dependent contact mechanics, specifically how microtexture and macrotexture interactions change with speed and how this correlates with real-world tire-pavement friction. The paper currently states that low speeds show stronger correlation with PSV but could further integrate tribological principles to explain the observed trends in greater depth.

Response Thank you again for your insightful feedback. We have deepened the theoretical basis and enriched the discussion by explicitly addressing speed-dependent contact mechanics in relation to micro- and macro-texture interactions. Specifically, we incorporated quantitative analysis of friction decay across speed regimes, connected our findings to tribological theory, and clarified how these dynamics relate to tire-pavement friction.

In Section 3.4.2, we now explain the observed non-linear speed effect by clarifying that at low speeds, micro-texture supports adhesive friction through more frequent tire–aggregate contacts, while at high speeds, macro-texture governs friction predominantly via hysteretic mechanisms. We also briefly discuss the initial friction peak in terms of asphalt film removal and exposure of aggregate micro-asperities (Lines 569–612).

In Section 3.4.3, we connect our findings to Persson’s contact mechanics framework to justify why the PSV–DF correlation is stronger at low speeds and weakens as speed increases—reflecting the shift from adhesive to hysteretic friction dominance (Lines 656–681).

These additions anchor our empirical observations in well-established friction theory, substantially improving the manuscript’s clarity and relevance to real-world tire–pavement interactions.

3- The results for MPD attenuation indicate that CB aggregates retain surface texture better than limestone. Yet, the study should clarify the relationship between MPD and skid resistance more explicitly, considering that MPD alone may not capture microtexture variations which critically affect friction. A combined analysis involving both macrotexture and microtexture measurements, possibly using advanced surface profilometry, would strengthen the interpretation of texture stability.

Response: Thank you for this insightful suggestion. We have deepened our discussion of texture stability by incorporating a new micro-texture analysis alongside the existing MPD results, thereby clarifying how surface morphology influences skid resistance beyond what MPD alone reveals.

In Section 3.4.1(Lines 516-534), we added Fig 8 (Line 522), which is a set of post-polishing surface topography images, and briefly describe how LS shows extensive smoothing while CB aggregates retain pronounced asperities. This addition directly links MPD attenuation to micro-texture degradation and highlights the role of preserved roughness (supported by higher PSV) in sustaining skid performance.

These changes integrate macro- and micro-texture insights to offer a more comprehensive understanding of texture stability and its real‑world frictional implications.

4- While the paper reports superior performance of 88# CB over 75# and 85# CB, it does not sufficiently explain the practical significance of these differences in field conditions. The authors should discuss whether the incremental improvement from 85# to 88# CB justifies potential cost or availability trade-offs. Moreover, the differences in chemical and physical properties between these grades could be linked more explicitly to their observed performance metrics to clarify material selection criteria.

Response: Thank you for this constructive comment. In response, we have revised Section 4.1 to explicitly link the chemical and mineralogical differences among CB grades (e.g., Al₂O₃ content, corundum/mullite ratio, Aggregate Hardness Parameter values) to their observed performance trends (e.g., PSV decay, DF stability). Additionally, we elaborated on the practical implications of these differences, highlighting how the superior long-term performance of 88#CB contributes to extended service life and reduced maintenance needs in safety-critical pavement sections.

Furthermore, in Section 4.2, we added a cost-benefit discussion comparing 88#CB and 85#CB under different application scenarios. This addition clarifies how the selection between CB grades can be guided by traffic volume, safety priorities, and budget constraints.

These revisions improve the practical relevance of the study by helping practitioners understand when the incremental benefits of higher-grade CB materials justify additional cost, thereby supporting more informed material selection in pavement design.

5- The study finds that limestone mixtures have marginally better water stability than CB mixtures due to chemical bonding effects. However, the explanation could benefit from further experimental validation, such as surface energy measurements or binder-aggregate adhesion tests, to corroborate this chemical bonding hypothesis. Additionally, given that the 88# CB mixture’s water stability is close to limestone, the impact of water stability on long-term skid resistance should be discussed in the context of field aging and freeze-thaw cycles.

Response: We appreciate the reviewer’s insightful suggestion. While this study primarily investigates the frictional and long-term wear performance of CB aggregates in SMA-7 mixtures, we agree that the mechanisms of water stability merit clarification. However, detailed investigations on binder–aggregate adhesion mechanisms (including surface energy testing and chemical bonding effects) have been comprehensively addressed in prior studies, as cited (Line 390, Reference [37]). Therefore, we referenced and interpreted these findings rather than conducting additional adhesion tests.

To address the reviewer’s concern, we revised the water stability explanation in Section 3.2 (Lines 386–394), incorporating prior findings to explain the roles of CaO-based chemical bonding in LS and the impact of Al₂O₃ purity and SiO₂ content on CB's adhesion characteristics. Additionally, we expanded Section 4.3 (Lines 830–837) to highlight the implications of water stability on long-term friction performance under environmental aging, suggesting directions for future work (e.g., freeze–thaw texture retention and field validation under climatic variability).

These changes strengthen the contextual interpretation of the results and clarify the scope and rationale of our current work while outlining potential extensions that can address durability-performance coupling in more depth.

Response to Reviewer 2

Reviewer #2:

The manuscript titled “Impact of Calcined Bauxite Aggregates on the Polishing Resistance and Skid Resistance Performance of SMA-7 Asphalt Mixtures” addresses a pertinent research topic within the realm of structural engineering. This study investigates how using calcined bauxite as a coarse aggregate affects the surface durability and safety characteristics—specifically polishing resistance and skid resistance—of a particular type of asphalt mixture (SMA-7). The research likely aims to determine whether this aggregate can improve road safety and extend pavement life under traffic wear. While the study presents an interesting perspective, a thorough evaluation highlights several areas requiring improvement. To enhance the manuscript's clarity, it is essential to refine the articulation of key concepts, ensuring that the arguments are logically structured and effectively communicated. Additionally, the methodological approach should be strengthened by providing a more detailed explanation of the research design, data analysis techniques, and validation processes to ensure scientific rigor. Furthermore, a deeper discussion of the findings, including their broader implications and potential applications, would significantly enhance the manuscript’s impact. Addressing these aspects will improve the overall coherence, credibility, and contribution of the study, aligning it with the expected scholarly standards.

Response: We sincerely thank the reviewer for the constructive overall feedback. In response, we have carefully revised the manuscript to improve clarity, strengthen methodological descriptions, and deepen the discussion of the findings. Specifically, we refined the expression of key concepts in the introduction, result and discussion sections to enhance logical flow and readability. We also provided additional details in the methodology section regarding material preparation and test procedures to ensure greater transparency and scientific rigor. Furthermore, we expanded the discussion to better highlight the practical implications of the results and the potential applications in pavement design and maintenance. We hope these improvements address the reviewer’s suggestions and contribute to a clearer and more impactful manuscript.

1- The introduction section should be improved. The literature review was not suitable.

Response: Thank you for the valuable feedback. We have substantially revised the introduction to clarify the research background and enhance the literature review (Lines 43-47, Lines 74-95). Specifically, we emphasized the multi-scale damage mechanisms governing friction degradation, the critical role of aggregate properties in skid resistance, and the dual-scale texture contributions of micro-roughness and macro-drainage structure. Additionally, we introduced the advantages of CB aggregates in resisting polishing and chemical degradation, providing stronger motivation for the study. Relevant recent studies have been cited to support these updates.

2- Why was the ASTM C33 classification curve of aggregates not presented?

Response: We appreciate this comment. The aggregate gradation curves have now been added to the manuscript (Fig 5, Line 360) to improve transparency regarding particle size distribution and classification.

3- The quality and detail of the SEM images were not suitable for publication.

Response: We agree and have replaced the SEM images with same resolution versions (Fig 1, Line 151) that more clearly reveal the surface morphology and texture characteristics of the aggregates.

4- More details should be provided about PSV specimens and tests.

Response: Thank you for the valuable suggestion. We have added further detail on the preparation (Lines 173-190) and conditioning of PSV specimens, and refined the illustration of the test process (Fig 2, Line 171) to improve clarity and reproducibility.

5- Why were no keywords not presented in the presented manuscript?

Response: Thank you for your observation. As per the PLOS ONE formatting guidelines, keywords are not included in the main manuscript file. However, we have provided the relevant keywords through the submission system as required. The keywords submitted are: Pavement skid resistance; Calcined bauxite; Long-term polishing resistance; Skid durability; Po

---

## [Decision Letter · Decision Letter 1]

Impact of Calcined Bauxite Aggregates on the Polishing Resistance and Skid Resistance Performance of SMA-7 Asphalt Mixtures

PONE-D-25-25733R1

Dear Dr. Gu,

We’re pleased to inform you that your manuscript has been judged scientifically suitable for publication and will be formally accepted for publication once it meets all outstanding technical requirements.

Kind regards,

Jiaolong Ren

Academic Editor

PLOS ONE

Additional Editor Comments (optional):

Reviewers' comments:

Reviewer's Responses to Questions

**Comments to the Author**

Reviewer #1: (No Response)

Reviewer #2: All comments have been addressed

2. Is the manuscript technically sound, and do the data support the conclusions?

Reviewer #1: (No Response)

Reviewer #2: Yes

3. Has the statistical analysis been performed appropriately and rigorously?

Reviewer #1: (No Response)

Reviewer #2: Yes

4. Have the authors made all data underlying the findings in their manuscript fully available?

Reviewer #1: (No Response)

Reviewer #2: Yes

5. Is the manuscript presented in an intelligible fashion and written in standard English?

Reviewer #1: (No Response)

Reviewer #2: Yes

Reviewer #1: The authors have successfully addressed all critical points raised in the previous review. The revisions enhance the clarity and robustness of the study's findings.

Reviewer #2: Comments have been addressed.

**Do you want your identity to be public for this peer review?** For information about this choice, including consent withdrawal, please see our Privacy Policy

Reviewer #1: No

Reviewer #2: No
